# OpenReview forum: "GLOV: Guided Large Language Models as Implicit Optimizers for Vision Language Models"
_TMLR — Accepted by TMLR_

### Review · Reviewer_UU4r · 2025-05-09

**Summary Of Contributions:**

This paper proposes a gradient-free prompt-tuning method for vision-language models (CLIP-based and autoregressive-based) using an LLM to propose example prompts, with a reward (or scoring) signal coming from the VLM. Specifically, the VLM is being steered by example prompts coming from an LLM (which the authors consider it to be a human-like model). The LLM proposes candidate prompts, which are then ranked by the VLM through a fitness function (here, accuracy) calculated on a small held-out 1-shot test set. The ranked prompts (along with the best-scoring and worst-scoring ones from previous optimization steps) are given to the LLM as form of in-context guidance examples and the LLM is steered to output prompts such that the VLM performs well on the given task. The authors also propose to guide the LLM through control vectors (calculated from the difference between positive and negative prompts) to further steer the LLM. Experiments are conducted on a wide set of classification dataset and VLM safety benchmarks and show promising results.

**Audience:**

Yes

**Broader Impact Concerns:**

No issues

**Claims And Evidence:**

Yes

**Requested Changes:**

In general I find this paper interesting and useful for the community. For me, the impressive improvements of LLava-like models and safety benchamrks, with an input-example-based and gradient-free optimization technique is enough. The authors can focus on addressing W1, W2, W3, and W4. The minor weaknesses can be addressed directly in the final manuscript.

**Strengths And Weaknesses:**

Strengths

- The idea of gradient-free LLM-guided optimization is interesting as it does not involve changing the base model, but rather automatic steering through examples at the input, which act as long-term memory (as if we use a huge batch size to update the paramters, but in a far more efficient way). This is more effective than soft prompt-tuning methods (e.g., COop) because the learned prompts are interpretable, rather than being dense uninterpretable vectors. Although this is not novel (previous works in Section 2.3 do something similar), I find the two aspects of (1) using the long-term history-buffer for guidance, and (2) showing the impressive performance of LLava-like models (which benefit hugely from prompt optimization) more important than the novelty aspect. Specifically, this work shows that the hand-crafted prompts used in [R1] (which [R1] claims to have no effect on the downstream task performance) are the actual problem, and by optimizing them we can significantly improve downstream task performance.
- Huge suit of datasets evaluated on, and impressive performance on autoregressive VLMs and safety VLM benchmarks. Optimizing for safety instructions is something I find very important in current large models.
- The optimization of general dataset-specific prompts rather than class-specific prompts as in previous works, is interesting.

Weaknesses:
- [W1] For steering the LLM with the offset vector, this will be limited to open-source models (which are still way behind OpenAI models and closed-source models). The method without steering would be more desirable because it could be applied to any LLM (even closed-source ones). Have the authors tried using closed-source models without offset vector steering, rather than low-performing Llama models? Furthermore, the idea of activation-space steering (commonly referred to as control/function vectors) is proposed in [R2], which I think deserves a citation.
- [W2] The performance on CLIP-like models on ImageNet is marginally better, and worse without activation guiding. I think this method is not effective for dual-encoder models.  This is not a weakness, but rather a feature, showing that we have reached a saturation in prompt-tuning methods for CLIP-based models. Can the authors justify why that might be the case? In my opinion, encoder-based models only contextually embed the prompt, but many fine-grained details in the prompt (e.g., negation) are lost, which might be the reason.
- [W3] The authors do not report the latency/speed. How long does it need to optimize the prompts? I find this important, especially for LLava-like models.
- [W4] Im not sure why the authors generate a description (along with the category) in LLava-like models and then measure the similarity of this generation with the ground-truth class with a text embedding model, as a way to measure accuracy. Why not prompt Llava to classify the image with a single or two-word response (given a possible pool of classes, like in [R1])? This is possible in all Llava models. Can the authors clarify their way of doing this?

Minor Weaknesses:
- In the example prompts in Figure 2, these are supposed to be general, for all classes. But here they seem to be for the specific class (cat).
- In Eq. 2, Shouldnt there also be a summation over all classes? l_c(x) should be the probability over all classes, not for a single class, but here it seems to be a scalar for a single class.

[R1] Why are Visually-Grounded Language Models Bad at Image Classification?, NeurIPS 2024\
[R2] Function Vectors in Large Language Models, ICLR 2024

---

> ### Author Response · Authors · 2025-05-22
> **Rebuttal Part 1**
>
> We sincerely thank the reviewer for the time and effort spent in reviewing our paper. Here, we address all the weaknesses mentioned. Please also note that we have updated the manuscript with the requested changes and the changes are reflected in teal color. For preparing the potential camera ready version, we will change the color to black for homogeneity.
>
> **Application of GLOV beyond Open-source LLMs and Missing Related Work.**
> We acknowledge the concern of the reviewer and indeed the guidance scheme can only be applied when the activations of the LLM are accessible. However, we would like to point out that in our paper we proposed two variants of our approach, i.e., with and without guidance. Our experiments show that our white-box approach also fares well compared to baselines. For example, our GLOV (without guidance) achieves $2.5\%$ and $19.0\%$ average improvements (over $16$ datasets) for the base CLIP and LLaVA One Vision models. Furthermore, our GLOV (without guidance) also outperforms the other white-box approach LLM-OPT by $1.7\%$ (averaged over $16$ datasets) for the CLIP models. These results highlight that our prompt optimization method can indeed be helpful in obtaining strong performance gains even when the activations of the LLM are not accessible.
>
> We also test our method (without applying the guidance scheme) with closed source GPT-4o and obtain the following results:
>
> | Method                            | Eurosat | INA  | Resisc | INR  | UCF101 |
> |----------------------------------|---------|------|--------|------|--------|
> | CLIP (VIT B/32)                             | 35.8    | 28.2 | 54.1   | 65.4 | 60.4   |
> | GLOV w/o guidance (llama)         | 46.9    | 31.2 | 61.4   | 67.8 | 63.0   |
> | GLOV w/o guidance (GPT-4o)   | **48.3**    | **31.6** | **62.2**   | **68.4** | **63.5** |
>
> We find that using a better LLM can indeed help our method to achieve higher accuracy gains. Please also note that we have also cited the related work [R1] - thank you for the suggestion and bringing it to our attention.
>
>
>
> **Performance Improvement for CLIP Models on ImageNet.**
> We appreciate the reviewer’s insightful observation.
> While it is true that improvements for CLIP-like models on ImageNet appear modest, we would like to highlight that our method achieves a $2.6\%$ gain over the baseline using the default prompt "*a photo of a {}*", and a $1.2\%$ gain even when compared to a strong handcrafted prompt set consisting of $80$ templates. We believe these gains are meaningful in the context of a challenging large-scale dataset like ImageNet, especially considering that our approach uses only three prompts discovered via GLOV.
>
> That said, we also agree with the reviewer’s broader point: prompt-tuning for CLIP-like models appears to be reaching saturation. This is further supported by comparisons with state-of-the-art prompt-tuning methods such as [1], which use more than 300,000 per-category descriptions to achieve $65.0\%$ top-1 accuracy, while our method achieves $64.5\%$ with just $3$ dataset-level prompts and no class-level customization.
>
> As for *why* this saturation occurs, we hypothesize, in agreement with the reviewer, that it could be due to limitations inherent to encoder-only architectures.
> These models might primarily contextualize the prompt as a whole rather than understanding fine-grained linguistic structures such as negation, nuance, or compositionality.
> Moreover, prior work [2, 3, 4] has shown that CLIP’s text encoder often behaves like a bag-of-words, representing input text as a collection of individual concepts rather than modeling their structural relationships.
> In contrast, decoder-based VLMs like LLaVA integrate powerful LLMs trained on massive natural language corpora. These models are better equipped to process and respond to subtle prompt variations, enabling richer understanding and more expressive outputs.
>
> **Latency/Speed for Optimizing Prompts.**
> Thank you for raising this important point. We agree that understanding the optimization cost is crucial for evaluating the practicality of prompt tuning methods.
>
> To address this, we report wall-clock times (on a single NVIDIA A40 GPU) for optimizing prompts across different numbers of ImageNet classes using LLaVA-One-Vision. While Table 4 in the main manuscript presents the performance under the sub-shot setting, the table below complements it by including the corresponding optimization times:
>
> |      Classes               | 1000      | 50     | 20     | 15     | 10   | SOURCE |
> |---------------------|---------|--------|--------|--------|--------|--------|
> | Results             | 51.7  |49.9  | 47.0  | 46.6 | 44.8   | 36.5   |
> | Wall-clock Time (hours, A40)     | ~72| ~4| ~3 | ~2.30  | ~1.50|    -    |
>
> --- continued ---

---

> > ### Author Response · Authors · 2025-05-22
> > **Rebuttal Part 2**
> >
> > Our findings indicate that optimizing prompts on all 1000 classes yields the highest performance (51.7%) but incurs the highest computational cost (~72 hours). However, a key insight from our experiments is that optimizing prompts on a subset of classes provides comparable gains with significantly reduced optimization time. For instance, using only 50 classes achieves 49.9% accuracy (just 1.8% below the full setting) while reducing optimization time to approximately 4 hours. This represents an over 18× speedup, greatly improving the method’s accessibility for large-scale models like LLaVA. Further cost reductions are possible by decreasing the number of classes involved in prompt optimization, since the primary bottleneck lies in evaluating the 1-shot samples at each optimization step.
> > In contrast, for CLIP-type models, where the vision encoder's outputs can be cached, and only the text embeddings have to be generated at each step, optimization is significantly faster. For example, full prompt optimization on ImageNet takes approximately 3 hours on a single NVIDIA A40 GPU.
> >
> > These results demonstrate that our method not only scales effectively with the number of classes but also enables a flexible trade-off between compute cost and performance, making it both efficient and practical for real-world deployment.
> >
> > **Evaluation Protocol.**
> > First, we note that the approach of using a sentence embedding model to compare the outputs of decoder-based VLMs with the ground-truth class names is widely adopted in recent literature [5, 6, 7]. This is especially important in generative VLMs like LLaVA, where the output is not a direct class label but rather a free-form textual response that often includes a description in addition to the class.
> > Second, we emphasize that our evaluation protocol leads to higher baseline accuracy than the prompting-based classification method used in [8], which uses two protocols: *closed-world* (model chooses from a list) and *open-world* (model generates freely). In fact, [8] shows that the *closed-world* setting often performs worse (see Table 1 in [8]).
> >
> > To further clarify, we report the results on the same four datasets evaluated in [8] using pre-trained LLaVA models. Our baseline, using a sentence embedding similarity method, outperforms both closed - and open-world settings in all cases:
> >
> > |                     | ImageNet | Oxford Flowers | Stanford Cars | Caltech101 |
> > |---------------------|----------|----------------|----------------|------------|
> > | Closed-world [8]    | N/A      | 16.1           | 3.6            | 77.3       |
> > | Open-world [8]      | 32.3     | 17.7           | 0.0            | 54.2       |
> > | Our Baseline        | **36.5** | **19.4**       | **21.7**       | **77.7**   |
> >
> > These results indicate that prompting the model to classify with only one or two words often yields suboptimal outputs, sometimes even degrading to 0% accuracy (as in Stanford Cars). This is likely due to the generative nature of these models, which can include additional phrasing or uncertainty.
> >
> > In contrast, using a sentence embedding model to compare the generated output with ground-truth class names makes evaluation more robust and accurate, especially when outputs are descriptive or verbose.
> >
> > **Minor Weaknesses.**
> >
> > 1. *Prompts in Fig.2:* Please note that these prompts are chosen from the best-performing prompts for the OxfordPets dataset. This dataset contains images of cats and dogs from $37$ distinct categories. This is the reason why these prompts might appear to be targeted towards the 'cat' category. The top-performing prompts for OxfordPets are provided in the appendix (Section E.6, Page 29). We have also updated the manuscript and added the information about the dataset (i.e., OxfordPets) for which these prompts are displayed in the caption of Fig.2. Please let us know if you would want any additional changes.
> >
> > 2. *Summation in Eq.2:* Please note that Equation 2 defines a fitness metric based on the model's top-1 accuracy across the dataset $\mathcal{D}$, using the zero-shot class probabilities defined in Equation 1. Since $l_c(x)$ is already a probability (from a softmax over all classes), $\arg\max_{c} l_c(x)$ correctly retrieves the predicted class. The summation in Equation 2 is over the dataset, not over the classes, and appropriately so for evaluating classification accuracy.

---

> > > ### Author Response · Authors · 2025-05-22
> > > **References**
> > >
> > > [1] Meta-Prompting for Automating Zero-shot Visual Recognition with LLMs, ECCV 2024.
> > >
> > > [2] Teaching Structured Vision-Language Concepts to Vision-Language Models, CVPR 2023.
> > >
> > > [3] Text Encoders Bottleneck Compositionality in Contrastive Vision-language Models, EMNLP 2023.
> > >
> > > [4] Mass-producing Failures of Multimodal Systems with Language Models, NeurIPS 2024.
> > >
> > > [5] Vocabulary-free Image Classification, NeurIPS 2023.
> > >
> > > [6] Democratizing Fine-grained Visual Recognition with Large Language Models, ICLR 2024.
> > >
> > > [7] Are Vision Language Models Texture or Shape Biased and Can We Steer Them?, ICLR 2025.
> > >
> > > [8] Why are Visually-Grounded Language Models Bad at Image Classification?, NeurIPS 2024

---

### Review · Reviewer_zCnW · 2025-05-20

**Summary Of Contributions:**

The paper introduces GLOV, a novel framework that leverages large language models (LLMs) as implicit optimizers to discover high-quality natural-language prompts for vision-language models (VLMs). At each iteration, GLOV queries an LLM (e.g., Llama-3) with in-context examples ranked by a downstream fitness function (e.g., zero-shot accuracy from CLIP or attack success rate for safety), and further steers generation by adding an offset vector to the LLM’s hidden state, where \(H^+\) and \(H^-\) are sentence embeddings of the best and second-best prompts. Without any gradient fine-tuning, GLOV achieves up to 15.0 % relative gain in classification accuracy for dual-encoder VLMs and 57.5 % gain for encoder-decoder models, and reduces attack success rates by up to 60.7 % on safety benchmarks.

**Audience:**

Yes

**Broader Impact Concerns:**

While GLOV can improve VLM performance and safety, it may amplify biases present in the LLM’s training data when generating prompts, leading to skewed downstream behavior.

**Claims And Evidence:**

Yes

**Requested Changes:**

- Typo: Fig.1 LlaVa One Vision -> LLaVA-One-Vision、
- Provide a detailed cost analysis (GPU hours) and discuss runtime trade-offs to clarify practical feasibility.
- Include experiments in a fully zero-shot setting or discuss strategies to mitigate reliance on labeled data (would strengthen).

**Strengths And Weaknesses:**

Strengths:
- Innovation: framing prompt search as an agentic feedback loop between an LLM and a VLM, with an efficient embedding-space guidance mechanism.  LLM-VLM interaction is interesting.
- Generality: applies to both contrastive dual-encoder and generative encoder-decoder VLMs and supports tasks from classification to safety.
- Evaluation is conduct on varied benchmarks to show the generalization ability.

Weaknesses:
1. Revise the related work section such as LLMs and VLMs, is quite boarder. One close area is prompt tuning for LLM and VLM, and author should discuss these works.
2. The Fig. 2 is hard to understand, should be refined and polished and make it clearly. Such as each letter denote what?
3. Why evaluate on Object recognition and VLM safety; Obj. recognition should be Image Recognition?
4. Robustness: The experiment benchmark is toy, it is not clear whether the reported results come from a single seed or are averaged over multiple runs with measures of spread (e.g., mean and variance).

---

> ### Author Response · Authors · 2025-05-22
> **Rebuttal Part 1**
>
> We sincerely thank the reviewer for the time and effort spent reviewing our paper. Here, we address all the weaknesses mentioned. Please also note that we have updated the manuscript with the requested changes, and the changes are reflected in teal color. For preparing the potential camera-ready version, we will change the color to black for homogeneity.
>
>
> **Revision of Related Work to Include Prompt Tuning Approaches.**
> Thank you for your suggestion. We have included detailed paragraphs that highlight the prompt tuning approaches in Natural Language Processing (NLP) and also in the domain of Vision-Language Models (VLMs). We are also open to making more changes (citing additional works) if required.
>
> **Revision of Fig. 2**
> Thank you for your suggestion. We have updated the Fig. 2 and included details on all the symbols. The updated version indeed makes the figure more comprehensible.
>
>
> **Evaluations on Object Object Recognition and VLM Safety.**
> We appreciate the reviewer’s comment and the opportunity to clarify. Object recognition, as evaluated in our work, refers to standard image recognition tasks on benchmarks like ImageNet, which are widely accepted as the de facto testbeds for evaluating visual understanding capabilities of Vision-Language Models (VLMs) and is also considered as one of the fundamental tasks in computer vision. These extensive experiments (on 16 widely accepted image recognition benchmarks) allow us to highlight the effectiveness of our prompt optimization method.
>
> In addition to image recognition, we also include evaluations on VLM safety benchmarks to demonstrate the robustness and broader applicability of our method. VLM safety is a critical concern, especially as VLMs are increasingly deployed in real-world applications. Our evaluations on two recent benchmarks highlight that our method is able to search for safety prompts that can greatly decrease the attack success rates (ASR) on VLMs.
> These results also show that safety alignment can be achieved without the need for explicit safety fine-tuning but only searching for optimal safety instructions, which has not been shown before in literature.
>
> To show further generalization of our method to open-ended tasks like Visual Question Answering (VQA), we evaluate our method on the large-scale SEED benchmark, we find that we can obtain a $2.1\%$ gain ($67.8$ --> $69.9$) for the LlaVA-OV model, by prepending the following prompt (optimized through GLOV) to the questions in the dataset:
>
> - *Which object or scene element in the image is the primary driver of the question's task or action, and how does it interact with the textual cues and context to elicit a specific answer?*
>
>
>
> **Robustness of Experimental Results.**
> The experiments listed in the paper are performed with a single seed. However, to test the statistical significance of our results, we report the mean and variance over $5$ independent runs in Table 11 (submitted manuscript, appendix).
> However, for ease of reference, we again list those results here, in the table below.
>
> |          Model | ImageNet | ImageNetA | ImageNetS | UCF101       |
> |---------------|----------|-----------|-----------|--------------|
> | CLIP (VIT B/32)          | 61.9     | 28.2      | 40.3      | 60.4         |
> | GLOV          | 64.3 ± 0.43 | 32 ± 0.69 | 43.1 ± 0.25 | 63.9 ± 0.34 |
>
> We find that the obtained results are indeed statistically significant.

---

> > ### Author Response · Authors · 2025-05-22
> > **Rebuttal Part 2**
> >
> > **Typo in Fig.1**
> > Thank you for pointing it out. The typo is fixed in the updated version of the manuscript.
> >
> > **Cost Analysis.**
> > Thank you for raising this important point. We agree that understanding the optimization cost is crucial for evaluating the practicality of prompt tuning methods.
> >
> > To address this, we report wall-clock times (on a single NVIDIA A40 GPU) for optimizing prompts across different numbers of ImageNet classes using LLaVA-One-Vision. While Table 4 in the main manuscript presents the performance under the sub-shot setting, the table below complements it by including the corresponding optimization times:
> >
> >
> > |      Classes               | 1000      | 50     | 20     | 15     | 10   | SOURCE |
> > |---------------------|---------|--------|--------|--------|--------|--------|
> > | Results             | 51.7  |49.9  | 47.0  | 46.6 | 44.8   | 36.5   |
> > | Wall-clock Time (hours, A40)     | ~72| ~4| ~3 | ~2.30  | ~1.50|    -    |
> >
> > Our findings indicate that optimizing prompts on all 1000 classes yields the highest performance (51.7%) but incurs the highest computational cost (~72 hours). However, a key insight from our experiments is that optimizing prompts on a subset of classes provides comparable gains with significantly reduced optimization time. For instance, using only 50 classes for optimization (50 labeled samples) achieves 49.9% accuracy (just 1.8% below the full setting) while reducing optimization time to approximately 4 hours. This represents an over 18× speedup, greatly improving the method’s accessibility for large-scale models like LLaVA. Further cost reductions are possible by decreasing the number of classes involved in prompt optimization, since the primary bottleneck lies in evaluating the 1-shot samples at each optimization step.
> > In contrast, for CLIP-type models, where the vision encoder's outputs can be cached, and only the text embeddings have to be generated at each step, optimization is significantly faster. For example, full prompt optimization on ImageNet takes approximately 3 hours on a single NVIDIA A40 GPU.
> >
> > These results demonstrate that our method not only scales effectively with the number of classes but also enables a flexible trade-off between compute cost and performance, making it both efficient and practical for real-world deployment.
> >
> >
> > **Discussion on Fully zero-shot Setting and non-reliance on Labeled Data.**
> > As the prompt search in GLOV is posed as an optimization problem, thus, it requires access to a small amount of labeled data to obtain the effectiveness of each prompt at each optimization step.
> > For the main experiments in Tables 1 & 2, we only use 1-shot labeled examples.
> > To further reduce the reliance on labeled data, we test our method in a *sub-shot* setting where we use labeled examples from only a small subset of classes from the dataset.
> > These results are listed as an ablation in Table 1, where we find that by optimizing the prompts only on $50$ labeled samples (from $50$ categories, instead of $1000$ in ImageNet), our method can still achieve an absolute improvement of $13.4\%$ as compared to the baseline, while only showing a degradation of $1.8\%$ when compared with optimizing the prompts on $1000$ samples from $1000$ classes.
> > These results are a step towards making the optimization method unsupervised.
> > In the future, our GLOV can potentially be made completely unsupervised (and zero-shot) by employing unsupervised metrics to measure the effectiveness of prompts.
> > These metrics could include entropy measurement [1], and choosing the prompts that decrease the overall entropy of the task.
> > Furthermore, we can also employ LLMs as a judge to provide a signal of effectiveness for the prompts, which could also completely lift the burden of data-labeling.
> > We have added this discussion as a paragraph at the end of the method section.
> >
> > [1] A Mathematical Theory of Communication.

---

> > > ### Author Response · Authors · 2025-05-23
> > > **Broader Impact**
> > >
> > > We acknowledge the broader impact concern raised by the reviewer, and we have added a statement at the end of the updated manuscript. We are also willing to modify it, if deemed necessary. Thank you!

---

### Review · Reviewer_oyCj · 2025-05-22

**Summary Of Contributions:**

This paper proposes GLOV (Guided Large Language Models as Implicit Optimizers), a method that uses Large Language Models (LLMs) to iteratively discover and refine natural language prompts for Vision-Language Models (VLMs), aiming to enhance performance on downstream vision tasks like object recognition and VLM safety. GLOV prompts an LLM with a task description and ranked in-context examples of previously generated VLM prompts (and their accuracies on a small training set). A key novelty is the explicit guidance of the LLM's generation process at each step by adding an offset vector—derived from embedding differences between previous effective (positive) and ineffective (negative) prompts—to an intermediate layer of the LLM. This biases the LLM towards generating language preferred by the target VLM. The authors evaluate GLOV on object recognition across 16 datasets using both dual-encoder (e.g., CLIP) and encoder-decoder (e.g., LLaVA) VLMs, and on VLM safety benchmarks, reporting significant performance improvements and reductions in attack success rates.

**Audience:**

Yes

**Broader Impact Concerns:**

Can this method possibly introduce certain bias in the prompt optimization process? We can clearly see a lot of vision-irrelevant words in the resulted prompt.

**Claims And Evidence:**

No

**Requested Changes:**

1. The LLM prompt includes system instructions, task descriptions, and ranked in-context examples from a "global history." Managing this global history and ensuring the LLM prompt doesn't become overly long or difficult for the LLM to process effectively, especially over many iterations, is challenging. It is important to provide more details about this part in the main paper and explicit show the trade-off between the baselines and the proposed method in terms of accuracy and efficiency.

2. Since the proposed method is claimed as an optimizer, the reviewer expects to see more fine-tuned baselines using either gradient descent or existing approaches of gradient free optimization methods. These are essential baselines to provide.

3. The current used VLMs are not strong enough. If they are already based on strong enough LLMs, would the proposed method still help?

**Strengths And Weaknesses:**

1. The proposed idea is interesting.
2. The paper is overall well presented.
3. The codebase could be useful for future research if released.

---

> ### Author Response · Authors · 2025-05-27
> **Rebuttal**
>
> We sincerely thank the reviewer for the time and effort spent reviewing our paper. Here, we address all the weaknesses/requested changes mentioned. Please also note that we have updated the manuscript with changes, and they are reflected in teal color. To prepare the potential camera-ready version, we will change the color to black for consistency.
>
> **Release of Codebase.**
> Please note that we have already attached the entire codebase as supplementary material. Further, as promised in the appendix, the codebase will also be publicly released upon acceptance.
>
> **Risk of LLM Prompts Becoming Overly Complicated.**
> We appreciate the reviewer’s observation regarding the potential complexity of managing the global history in the LLM prompt during iterative optimization. We would like to clarify that in our approach, the LLM prompt is carefully controlled to avoid excessive length or complexity.
>
> As detailed in Section 3.2 (*In-context examples* paragraph) of the main manuscript, at each optimization iteration, we only include a *subset* of the ranked examples, specifically, the top-*k* and bottom-*k* performing prompts (with *k* = 5, based on classification accuracy). This design ensures that the LLM prompt remains concise and computationally manageable across iterations. It also avoids the accumulation of all historical examples in the prompt and makes it more focused on identifying which language patterns are most effective for the downstream VLM.
> Due to space constraints, we provided a representative illustration of the LLM prompt in Appendix Figure 5.
>
> Please let us know if any further clarification is required on this point.
>
>
> **Gradient-based and Gradient-free Baselines.**
> We would like to clarify that we already compare with LLM-OPT [1] in Table 1 of the main manuscript. LLM-OPT is a recent method that employs LLMs in an iterative optimization scheme. We find that our proposed method GLOV outperforms LLM-OPT across all benchmarks evaluated in our work.
>
> Additionally, LLM-OPT has limited applicability to encoder-decoder Vision-Language Models, as it requires pre-defined VLM prompts. Such prompts are typically available only for CLIP-like dual encoders [2], and not for decoder-based models.
>
> To address the comparison with gradient-based parameter-efficient fine-tuning (PEFT) methods, we include results with gradient-based baselines in Table 10 of the Appendix (discussion in Section B.4), evaluating their performance in the 1-shot learning regime. For the reviewer’s convenience, we reproduce the table here:
>
> | Method           | ImageNet | ImageNetA | ImageNetS | UCF101 |
> |------------------|----------|-----------|-----------|--------|
> | CLIP             | 61.9     | 28.2      | 40.3      | 60.4   |
> | CoOp [3]         | 60.6     | 24.5      | 39.9      | 63.8   |
> | LORA (all)       | 59.9     | 27.1      | 37.1      | 58.2   |
> | LORA (attention) | 62.6     | 30.1      | 40.5      | 62.1   |
> | **GLOV**         | **64.5** | **32.5**  | **43.0**  | **63.8** |
>
> As shown, GLOV consistently outperforms gradient-based PEFT methods in extremely low-shot regimes. We hypothesize that this advantage arises because PEFT methods are more prone to overfitting when only a tiny amount of training data is available. In contrast, GLOV is gradient-free and thus inherently more robust to overfitting in this regime.
>
>
> **VLMs Based on Stronger LLMs in the Future.**
> In our study, we have already evaluated multiple strong VLMs, including various CLIP variants as well as decoder-based models such as LLaVA-One-Vision and Molmo, both of which are grounded in powerful LLMs.
> Our proposed method, GLOV, consistently improves performance across all these models.
>
> Interestingly, we observe that the improvements are substantially larger for decoder-based VLMs (up to 57.5%) compared to 15.0% for CLIP-like dual-encoder models.
> This suggests that GLOV-optimized prompts are particularly beneficial for VLMs built on strong LLMs, likely because such models better capture subtle linguistic patterns and structure [4, 5, 6].
>
> These findings indicate that as LLMs continue to advance, GLOV stands to benefit even more, since future models are expected to have an even deeper understanding of language, making them more responsive to optimized prompt structures.
>
> **Broader Impact Concern.**
> We acknowledge the broader impact concern raised by the reviewer, and we have added a statement at the end of the updated manuscript.
> We are also willing to modify it, if deemed necessary.
> Thank you!
>
> [1] Language Models as Black-Box Optimizers for Vision-Language Models
>
> [2] Learning Transferable Visual Models From Natural Language Supervision
>
> [3] Learning to Prompt for Vision-Language Models
>
> [4] Teaching Structured Vision-Language Concepts to Vision-Language Models, CVPR 2023.
>
> [5] Text Encoders Bottleneck Compositionality in Contrastive Vision-language Models, EMNLP 2023.
>
> [6] Mass-producing Failures of Multimodal Systems with Language Models, NeurIPS 2024.

---

### Decision · Action_Editor_MhWj · 2025-07-30

**Recommendation:** Accept with minor revision

**Additional Comments:**

As mentioned above, I request that the authors revise the manuscript by addressing the reviewers’ comments. Specifically, please ensure that all experiments and related discussions are included in the final version. Additionally, please strengthen the evaluation by comparing your method with more recent gradient-descent-free approaches and stronger baselines.

**Audience:**

Yes

**Audience Explanation:**

Enhancing the performance of VLMs on downstream tasks remains an important challenge, and the current work proposes a promising approach to address this issue. As such, the paper has the potential to make an impact on a broader research community.

**Claims And Evidence:**

Yes

**Claims Explanation:**

The reviewers found the proposed method to be interesting.
However, as they pointed out, there are still concerns regarding the experimental section.

Therefore, I request that the authors revise the manuscript by addressing the reviewers’ comments. Specifically, please ensure that all experiments and related discussions are included in the final version. Additionally, please strengthen the evaluation by comparing your method with more recent gradient-descent-free approaches and stronger baselines.